# ADAPTIVE PARAMETRIC PROTOTYPE LEARNING FOR CROSS-DOMAIN FEW-SHOT CLASSIFICATION

## ABSTRACT

Cross-domain few-shot classification induces a much more challenging problem than its in-domain counterpart due to the existence of domain shifts between the training and test tasks. In this paper, we develop a novel Adaptive Parametric Prototype Learning (APPL) method under the meta-learning convention for cross-domain few-shot classification. Different from existing prototypical few-shot methods that use the averages of support instances to calculate the class prototypes, we propose to learn class prototypes from the concatenated features of the support set in a parametric fashion and meta-learn the model by enforcing prototype-based regularization on the query set. In addition, we fine-tune the model in the target domain in a transductive manner using a weighted-moving-average self-training approach on the query instances. We conduct experiments on multiple cross-domain few-shot benchmark datasets. The empirical results demonstrate that APPL yields superior performance than many state-of-the-art cross-domain few-shot learning methods.

## 1 INTRODUCTION

Benefiting from the development of deep neural networks, significant advancement has been achieved on image classification with large amounts of annotated data. However, obtaining large amounts of annotated data is time-consuming and labour-intensive, while it is difficult to generalize trained models to new categories of data. As a solution, few-shot learning (FSL) has been proposed to classify instances from unseen classes using only a few labeled instances. FSL methods usually use a base dataset with labeled images to train a prediction model in the training phase. The model is then fine-tuned on the prediction task of novel categories with a few labeled instances (i.e. support set), and finally evaluated on the test data (i.e. query set) from the same novel categories in the testing phase. FSL has been widely studied in the in-domain settings where the training and test tasks are from the same domain (Finn et al., 2017; Snell et al., 2017; Lee et al., 2019). However, when the training and test tasks are in different domains, it poses a much more challenging cross-domain few-shot learning problem than its in-domain counterpart due to the domain shift problem.

Recently, several methods have made progress to address cross-domain few-shot learning, including the ones based on data augmentation, data generation (Wang & Deng, 2021; Yeh et al., 2020; Islam et al., 2021) and self-supervised learning (Phoo & Hariharan, 2020) techniques. However, such data generation and augmentation methods increase the computational cost and cannot scale well to scenarios with higher-shots (Wang & Deng, 2021). Some other works either require large amounts of labeled data from multiple source domains (Hu et al., 2022) or the availability of substantial unlabeled data from the target domain during the source training phase (Phoo & Hariharan, 2020; Islam et al., 2021; Yao, 2021). Such requirements are hard to meet and hence hamper their applicability in many domains. Although some existing prototypical-based few-shot methods have also been applied to address cross-domain few-shot learning due to their simplicity and computational efficiency (Snell et al., 2017; Satorras & Estrach, 2018), these standard methods lack sufficient capacity in handing large cross-domain shifts and adapting to target domains.

In this paper, we propose a novel Adaptive Parametric Prototype Learning (APPL) method under the meta-learning convention for cross-domain few-shot image classification. APPL introduces a parametric prototype calculator network (PCN) to learn class prototypes from concatenated feature vectors of the support instances by ensuring the inter-class discriminability and intra-class cohe-

sion with prototype regularization losses. The PCN is meta-learned on the source domain using the labeled query instances. In the target domain, we deploy a weighted-moving-average (WMA) self-training approach to leverage the unlabeled query instances to fine-tune the prototype-based prediction model in a transductive manner. With PCN and prototype regularizations, the proposed method is expected to have better generalization capacity in learning class prototypes in the feature embedding space, and hence effectively mitigate the domain shift and adapt to the target domain with WMA self-training. Comprehensive experiments are conducted on eight cross-domain few-shot learning benchmark datasets. The empirical results demonstrate the efficacy of the proposed APPL for cross-domain few-shot classification, by comparing with existing state-of-the-art methods.

The contributions of our proposed method are as follows:

1. We propose a novel adaptive prototype calculator network called Prototype Calculator Network (PCN). Our key contribution is that we use a parameterization mechanism to generate more representative prototypes and propose two loss functions to learn discriminative class prototypes by enforcing both inter-class discriminability and intra-class cohesion in the extracted feature space.

2. We propose a WMA self-training strategy that is tailored for the CDFSL problem. Compared to existing methods, it overcomes the barrier of requiring large amounts of additional data for the target domain and reduces domain shift by generating better pseudo-labels. It ensures that the produced pseudo-labels are stable and clean (not noisy) by jointly employing three mechanisms: Weighted-Moving-Average updating of prediction vectors, a rectified annealing schedule for the WMA and selectively sampling only the confident pseudo-labels to adapt the model.

3. Our proposed method work outperforms existing methods on both low-shot (5-shot) and high-shot (20-shot and 50-shot) classification tasks.

## 2 RELATED WORKS

### 2.1 FEW-SHOT LEARNING

Most FSL studies have focused on the in-domain settings. The FSL approaches can be grouped into three main categories: metric-based and meta-learning approaches (Finn et al., 2017; Snell et al., 2017; Lee et al., 2019), transfer learning approaches (Guo et al., 2019; Jeong & Kim, 2020; Ge & Yu, 2017; Yosinski et al., 2014; Dhillon et al., 2019) and augmentation and generative approaches (Zhang et al., 2018; Lim et al., 2019; Hariharan & Girshick, 2017; Schwartz et al., 2018; Reed et al., 2018). In particular, the representative meta-learning approach, MAML (Finn et al., 2017), learns good initialization parameters from various source tasks that make the model easy to adapt to new tasks. The non-parametric metric-based approach, MatchingNet (Vinyals et al., 2016), employs attention and memory in order to train a network that learns from few labeled samples. ProtoNet (Snell et al., 2017) learns a metric space where each class is represented by the average of the available support instances and classifies query instances based on their distances to the class prototypes. A few meta-learning works, such as RelationNet (Sung et al., 2018), GNN (Satorras & Estrach, 2018) and Transductive Propagation Network (TPN) (Liu et al., 2019), exploit the similarities between support and query instances to classify the query instances. MetaOpt uses meta-learning to train a feature encoder that obtains discriminative features for a linear classifier (Lee et al., 2019). Transfer learning methods initially train a model on base tasks and then use various fine-tuning methods to adapt the model to novel tasks (Guo et al., 2019; Jeong & Kim, 2020; Ge & Yu, 2017; Yosinski et al., 2014; Dhillon et al., 2019). Generative and augmentation approaches generate additional samples to increase the size of available data during training (Zhang et al., 2018; Lim et al., 2019; Hariharan & Girshick, 2017; Schwartz et al., 2018; Reed et al., 2018).

### 2.2 CROSS-DOMAIN FEW-SHOT LEARNING

Recently cross-domain few-shot learning (CDFSL) has started receiving more attentions (Guo et al., 2020; Phoo & Hariharan, 2020). Tseng et al. (2020) propose a feature-wise transformation (FWT) layer that is used jointly with standard few-shot learning methods for cross-domain few-shot learning. The FWT layer uses affine transformations to augment the learned features in order to help

the trained network generalize across domains. Du et al. (2022) address the domain shift problem by proposing a prototype-based Hierarchical Variational neural Memory framework (HVM), where the hierarchical prototypes and the memory are both learned using variational inference. Adler et al. (2020) propose a Cross-domain Hebbian Ensemble Fusion (CHEF) method, which applies an ensemble of Hebbian learners on different layers of the neural network to obtain a representation fusion. Data augmentation and data generation methods have also been used to bridge the gap between the source and target domains (Wang & Deng, 2021; Yeh et al., 2020; Islam et al., 2021). Wang & Deng (2021) propose an Adversarial Task Augmentation approach (ATA) to generate difficult training tasks in an adversarial fashion and improve the generalizability of few-shot methods across largely different domains. Islam et al. (2021) employ dynamic distillation and consistency regularization to train student and teacher network jointly on the source domain data and unlabeled data from the target domain. Hu et al. (2022) propose domain-switch learning framework with multiple source domains, and use re-weighted cross-entropy loss and binary KL divergence loss to prevent overfitting and catastrophic forgetting. Sun et al. (2021) adapt the explanation method of Layer-wise Relevance Propagation (LRP) to the FSL setup, which guides the FSL training by dynamically highlighting the discriminative features of the input samples.

Some other works (Triantafillou et al., 2019; Liu et al., 2021; Bateni et al., 2020; Doersch et al., 2020) have tested alternative cross-domain few-shot learning settings such as Meta-Dataset (Triantafillou et al., 2019) CDFSL setting where models are trained on several source-domain datasets and tested on multiple target-domain datasets. In this work, we focus on the CDFSL setting in Guo et al. (2020) as it is the most widely studied CDFSL setting.

## 3 Approach

### 3.1 Preliminary

The cross-domain few-shot learning problem aims to train a model on the source domain with its large set of labelled instances and then adapt the model to address the prediction task in the target domain with few labeled instances. We assume the two domains have different distributions in the input space ($\mathcal{P}_s \neq \mathcal{P}_t$) and have disjoint classes ($\mathcal{Y}_s \cap \mathcal{Y}_t = \emptyset$). In the target domain, the model is provided with a support set $S = \{(x_i, y_i)\}_{i=1}^{N_s}$ and tested on a query set $Q = \{(x_i, y_i)\}_{i=1}^{N_q}$ where $N_s$ and $N_q$ are the sizes of the support and query sets respectively. The support set is made up of $N$ classes with $K$ instances in each class, which is commonly described as N-way K-shot.

In the classic prototypical few-shot learning (Snell et al., 2017), each image $x$ first goes through a feature encoder $f_\theta$ and obtains its embedding vector in the feature space. Then, for each class in the support set, a prototype $p_n \in \mathbb{R}^D$ is computed as the average embedding vector of the support instances: $p_n = \frac{1}{K} \sum_{(x,y) \in S_n} f_\theta(x)$, where $S_n$ denotes the set of $K$ support instances from class $n$. To classify the query instances, the distances between each query sample and the prototypes of all classes in the support set are computed. Then the softmax function is used to normalize the calculated distances to obtain the class prediction probabilities as follows:

$$P(y = j | x) = \frac{\exp(-d(f_\theta(x), p_j))}{\sum_{n=1}^{N} \exp(-d(f_\theta(x), p_n))}, \tag{1}$$

where $d(.,.)$ is a distance function and $P(y = j | x)$ is the predicted probability that query sample $x$ belonging to class $j$. During the meta-training phase, the model is trained to minimize the cross-entropy loss on the query instances:

$$\mathcal{L}_{CE}(Q) = \sum_{x \in Q} \ell_{CE}(P_x, Y_x), \tag{2}$$

where $\ell_{CE}$ is the cross-entropy function, $P_x$ and $Y_x$ are the predicted class probability vector and ground-truth label indicator vector respectively for a query sample $x$.

### 3.2 Adaptive Parametric Prototype Learning

In this section, we present our proposed Adaptive Parametric Prototype Learning (APPL) method for cross-domain few-shot image classification. The overall framework of APPL is illustrated in

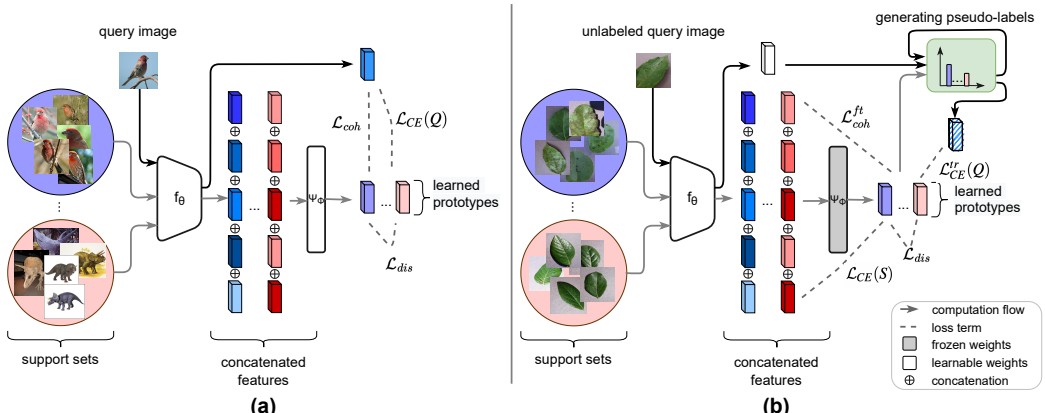

Figure 1: The proposed APPL method. **(a)** **Training on the source domain**. The concatenated feature vectors of each class are fed into the PCN ($\psi_\phi$) to produce class prototypes, which are used to compute the meta-training loss terms on query instances. **(b)** **Fine-tuning on the target domain.** Both the labelled support set and the unlabeled query set with soft pseudo-labels computed using WMA are used to fine-tune the feature encoder in the target domain with prototype based losses.

Figure 1. APPL first performs meta-training in the label-rich source domain by meta learning an adaptive prototype calculator network (PCN) after the feature encoder. PCN generates the prototype of each class by segregating information from the concatenated feature vectors of the K-shot support instances. Then the trained model can be fine-tuned for the few-shot classification task in the target domain using a weighted-moving-average (WMA) self-training approach, which aims to adapt the model to the target domain and further improve the quality of the learned class prototypes. We describe the details below.

### 3.2.1 ADAPTIVE PROTOTYPE CALCULATOR NETWORK

Simply averaging the support instances to calculate the class prototypes has the evident drawback of ignoring the inter-class and intra-class instance relations. To overcome this drawback, we propose to learn class prototypes from the support instances through a parametric prototype calculator network by enforcing both inter-class discriminability and intra-class cohesion in the extracted feature space. Such a parametric prototype generation mechanism is expected to produce more representative class prototypes from various support instance layouts, and guide the feature encoder to better adapt to the target domain through fine-tuning in the testing phase.

We define the adaptive prototype calculator network (PCN) as $\psi_\phi : \mathbb{R}^{K \cdot D} \to \mathbb{R}^D$, where $D$ is the size of the learned embeddings by the feature encoder, $f_\theta$, $K$ is the number of support instances per class, and $\phi$ denotes the parameters of the PCN. Specifically, PCN takes the concatenated feature vectors of the support instances of a given class as input, and outputs the prototype of the corresponding class:

$$p_n = \psi_\phi(concat(f_\theta(x_1^n), .., f_\theta(x_K^n))), \tag{3}$$

where $x_j^n$ denotes the $j$-th support instance from class $n$ and $p_n$ is the learned prototype of class $n$. By feeding the support instances of each class to $\psi_\phi$, we can obtain the prototypes for all the $N$ classes: $\mathbb{P} = \{p_1, p_2, .., p_N\}$.

We train the PCN during the meta-training phase using the few-shot training tasks in the source domain. Specifically, given the feature encoder trained on the support instances, we update the parameters of the PCN by minimizing the cross-entropy loss on the query instances, $\mathcal{L}_{CE}(Q)$. Moreover, we introduce two auxiliary regularization loss terms, a prototype discriminative loss and a prototype cohesive loss, to ensure the learned prototypes are both discriminative and representative of the underlying classes. To elaborate, the prototype discriminative loss $\mathcal{L}_{dis}$ aims to push the prototypes of different classes away from each other and is defined as follows:

$$\mathcal{L}_{dis} = \frac{1}{\sum_{\{p_i, p_j\} \in \mathbb{P}} d(p_i, p_j)}, \tag{4}$$

We in particular use a squared Euclidean distance as $d(\cdot, \cdot)$. By contrast, the prototype cohesive loss $\mathcal{L}_{coh}$ is designed to pull the prototypes and the query instances of their corresponding classes to be closer to each other:

$$\mathcal{L}_{coh} = \sum_{n=1}^{N} \sum_{x \in Q_n} d(p_n, f_\theta(x)), \tag{5}$$

where $Q_n$ denotes the set of query instances from class $n$. Overall, PCN is meta-trained in the source domain by minimizing the following joint loss:

$$\min_{\phi} \mathcal{L}_{train} = \mathcal{L}_{CE}(Q) + \lambda_{dis}\mathcal{L}_{dis} + \lambda_{coh}\mathcal{L}_{coh}, \tag{6}$$

where $\lambda_{dis}$ and $\lambda_{coh}$ are the trade-off hyper-parameters that control the contribution of the two regularization loss terms, $\mathcal{L}_{dis}$ and $\mathcal{L}_{coh}$, respectively.

### 3.2.2 WEIGHTED MOVING AVERAGE SELF-TRAINING

For cross-domain few-shot image classification, significant distribution discrepancies in the input image space typically exist between the source and target domains. Hence after meta-training the feature encoder $f_\theta$ and the PCN $\psi_\phi$ in the source domain, it is critical to fine-tune the feature encoder $f_\theta$ on the few-shot test task in the target domain to overcome the cross-domain gap as well as adapt $f_\theta$ to the target test task. Due to the scarcity of the labeled support instances in the target task, we propose to employ the unlabeled query instances with predicted soft pseudo-labels to increase the size and diversity of the target data for fine-tuning and mitigate the domain shift between the source and target domains. To this end, we develop a weighted-moving-average (WMA) self-training approach to compute the soft pseudo-labels and deploy the query instances for fine-tuning.

Specifically, at each iteration $i$ of the fine-tuning process, we first calculate the distances between each query instance $x$ and the class prototypes $[p_1^i, p_2^i, \cdots, p_N^i]$ produced by the PCN $\psi_\phi$ from the support set $S$ for all the $N$ classes, and form the following distance vector for $x$:

$$h^i(x) = [d(f_{\theta^i}(x), p_1^i), d(f_{\theta^i}(x), p_2^i), .., d(f_{\theta^i}(x), p_N^i)]^\top. \tag{7}$$

Then we use this distance vector $h^i(x)$ to perform weighted moving average update and maintain a weighted-moving-average distance vector $\tilde{h}^i(x)$ for the current iteration $i$ as follows:

$$\tilde{h}^i(x) = \alpha_i \, h^i(x) + (1 - \alpha_i) \, \tilde{h}^{i-1}(x), \tag{8}$$

where $\alpha_i$ is a trade-off parameter that controls the combination weights between distances computed from the current iteration and previous iterations. The weighted-moving-average distance vectors can then be used to compute the class prediction probabilities over each query instance $x$ by using the softmax function:

$$\tilde{P}^i(y = j|x) = \frac{\exp(-\tilde{h}_j^i(x))}{\sum_{n=1}^{N} \exp(-\tilde{h}_n^i(x))}, \tag{9}$$

where $\tilde{P}^i(y = j|x)$ is the probability of the query instance $x$ being assigned to class $j$ at iteration $i$. By using the predicted class probabilities as soft pseudo-labels, the query instances can subsequently be used to support the fine-tuning of $f_\theta$ in a self-training manner. The weighted-moving-average (WMA) update mechanism can stabilize the self-training process and dampen possible oscillating predictions for challenging query instances. Moreover, to increase the stability and convergence property of the WMA self-training, we adopt the following rectified annealing schedule for the WMA hyper-parameter $\alpha_i$:

$$\alpha_i = max(\alpha_{\min}, \gamma \, \alpha_{i-1}), \tag{10}$$

where $\alpha_{\min}$ specifies the smallest value $\alpha_i$ can take, and $\gamma \in (0, 1)$ is a reduction ratio parameter for updating the $\alpha$ value with increasing iterations. This annealing schedule can enable larger updates to the $\tilde{h}^i$ vectors in the beginning iterations of fine-tuning by starting with a large value $\alpha_0$, while gradually reducing the degree of update with the decreasing of $\alpha_i$ in later iterations.

With the soft pseudo-labels predicted by the current prototype-based model $(\theta^i, \phi)$, the query instances can be deployed through a cross-entropy loss to guide the further update, i.e., fine-tuning, of the feature encoder $f_\theta$. However, using all pseudo-labeled query instances may lead to noisy updates and negatively impact the model due to the low-confidence predictions of the pseudo-labels.

---

**Algorithm 1** Fine-tuning Procedure on Target Domain

---

**Input**: Target N-way-K-shot test task $(S, Q)$; source trained model $(f_\theta, \psi_\phi)$;
       hyper-parameters $\lambda_{dis}, \lambda_{coh}, \alpha_{min}, \alpha_0, \gamma, \epsilon$; initialize: $\theta^1 = \theta, \{\tilde{h}^0(x) = 0, \forall x \in Q\}$
**Output**: Fine-tuned feature encoder parameter $\theta$
**for** i = 1 **to** maxiters **do**
    Compute $p_n = \psi_\phi(concat(f_\theta(x_1^n), .., f_\theta(x_K^n)))$, for $S_n = \{x_1^n, \cdots, x_K^n\}, \forall n \in \{1, \cdots, N\}$
    $\nabla_\theta \mathcal{L}_{ft} \leftarrow 0$
    Compute $\alpha_i$ using Eq.(10)
    **for** $x \in Q$ **do**
        Compute the prediction probabilities $\tilde{P}^i(Y|x)$ using Eq.(7)(8)(9).
        $\nabla_\theta \mathcal{L}_{ft} \leftarrow \nabla_\theta \mathcal{L}_{ft} + \nabla_\theta \mathcal{L}_{CE}^{tr}((x, \tilde{P}^i(Y|x)); \theta, \phi)$
    **end for**
    $\nabla_\theta \mathcal{L}_{ft} \leftarrow \nabla_\theta \mathcal{L}_{ft} + \nabla_\theta \mathcal{L}_{CE}(S) + \lambda_{dis} \nabla_\theta \mathcal{L}_{dis} + \lambda_{coh} \nabla_\theta \mathcal{L}_{coh}^{ft}$
    $\theta^{i+1} \leftarrow \theta^i - \eta \nabla_{\theta = \theta^i} \mathcal{L}_{ft}; \quad \theta = \theta^{i+1}$
**end for**

---

Therefore, we choose to only employ query instances with high prediction confidence scores that are larger than a predefined threshold $\epsilon$ and compute the query-based cross-entropy loss as follows:

$$\mathcal{L}_{CE}^{tr}(Q) = \sum_{x \in Q} \begin{cases} \mathcal{L}_{CE}(x, \tilde{P}^i(Y|x); \theta, \phi) & \text{if } \max(\tilde{P}^i(Y|x)) > \epsilon, \\ 0 & \text{otherwise} \end{cases} \tag{11}$$

where $\tilde{P}^i(Y|x)$ denotes the soft pseudo-label vector computed via Eq.(9) for query instance $x$, while the maximum predicted probability, $\max(\tilde{P}^i(Y|x))$, is used as the prediction confidence score for query instance $x$. Here $\mathcal{L}_{CE}(x, \tilde{P}^i(Y|x); \theta, \phi)$ denotes the cross-entropy loss computed over the soft pseudo-labeled pair $(x, \tilde{P}^i(Y|x))$ with $f_\theta$ and $\psi_\phi$.

In addition to the cross-entropy loss on both the support and query instances, we also use the prototype regularization losses, $\mathcal{L}_{dis}$ and $\mathcal{L}_{coh}$, introduced in the meta-training phase to guide the fine-tuning process. Since the true labels of the query instances are unknown in the meta-testing phase, we modify the prototype cohesive loss $\mathcal{L}_{coh}$ and compute it on the support instances instead:

$$\mathcal{L}_{coh}^{ft} = \sum_{n=1}^{N} \sum_{x \in S_n} d(p_n, f_\theta(x)), \tag{12}$$

where $S_n$ is the set of support instances from class n. Overall, the feature encoder is fine-tuned by minimizing the following joint loss function with gradient descent:

$$\min_\theta \mathcal{L}_{ft} = \mathcal{L}_{CE}(S) + \mathcal{L}_{CE}^{tr}(Q) + \lambda_{dis}\mathcal{L}_{dis} + \lambda_{coh}\mathcal{L}_{coh}^{ft}. \tag{13}$$

This fine-tuning procedure is also summarized in Algorithm 1.

## 4 EXPERIMENTS

### 4.1 EXPERIMENTAL SETUP

**Datasets** We conducted comprehensive experiments on eight cross-domain few-shot learning (CDFSL) benchmark datasets. We use MiniImageNet (Vinyals et al., 2016) as the single source domain dataset, and use the following eight datasets as the target domain datasets: CropDiseases (Mohanty et al., 2016), EuroSAT Helber et al. (2019), ISIC (Tschandl et al., 2018), ChestX (Wang et al., 2017), CUB (Wah et al., 2011), Cars (Krause et al., 2013), Places (Zhou et al., 2017) and Planate (Van Horn et al., 2018). We use the same train/val/test split as Guo et al. (2020). We select the hyperparameters based on the test accuracy on the MiniImageNet validation set.

**Implementation Details** We use ResNet10 (He et al., 2016) as our backbone network and use a simple network made up of a single linear layer followed by ReLU activation to represent PCN. We train our prototype-based prediction model (feature encoder and PCN) on the source domain for 400

Table 1: Mean classification accuracy (95% confidence interval in brackets) for cross-domain 5-way 5-shot classification. [*] and [†] denote the results reported in Guo et al. (2020) and Wang & Deng (2021) respectively. Transductive methods are indicated using (T). Methods sharing query data via Batch Normalization are indicated using (BN)

| | ChestX | CropDisea. | ISIC | EuroSAT | Places | Planate | Cars | CUB |
|---|---|---|---|---|---|---|---|---|
| MatchingNet[*](Vinyals et al., 2016) | $22.40_{(0.7)}$ | $66.39_{(0.78)}$ | $36.74_{(0.53)}$ | $64.45_{(0.63)}$ | – | – | – | – |
| MAML(BN)[*](Finn et al., 2017) | $23.48_{(0.96)}$ | $78.05_{(0.68)}$ | $40.13_{(0.58)}$ | $71.70_{(0.72)}$ | – | – | – | – |
| ProtoNet[*](Snell et al., 2017) | $24.05_{(1.01)}$ | $79.72_{(0.67)}$ | $39.57_{(0.57)}$ | $73.29_{(0.71)}$ | $58.54_{(0.68)}$ | $46.80_{(0.65)}$ | $41.74_{(0.72)}$ | $55.51_{(0.68)}$ |
| MetaOpt[*](Lee et al., 2019) | $22.53_{(0.91)}$ | $68.41_{(0.73)}$ | $36.28_{(0.50)}$ | $64.44_{(0.73)}$ | – | – | – | – |
| RelationNet(BN)[†](Sung et al., 2018) | $24.07_{(0.20)}$ | $72.86_{(0.40)}$ | $38.60_{(0.30)}$ | $65.56_{(0.40)}$ | $64.25_{(0.40)}$ | $42.71_{(0.30)}$ | $40.46_{(0.40)}$ | $56.77_{(0.40)}$ |
| GNN[†](Satorras & Estrach, 2018) | $23.87_{(0.20)}$ | $83.12_{(0.40)}$ | $42.54_{(0.40)}$ | $78.69_{(0.40)}$ | $70.91_{(0.50)}$ | $48.51_{(0.40)}$ | $43.70_{(0.40)}$ | $62.87_{(0.50)}$ |
| TPN(T)[†](Liu et al., 2019) | $22.17_{(0.20)}$ | $81.91_{(0.50)}$ | $45.66_{(0.30)}$ | $77.22_{(0.40)}$ | $71.39_{(0.40)}$ | $50.96_{(0.40)}$ | $44.54_{(0.40)}$ | $63.52_{(0.40)}$ |
| MatchingNet+FWT[*](Tseng et al., 2020) | $21.26_{(0.31)}$ | $62.74_{(0.90)}$ | $30.40_{(0.48)}$ | $56.04_{(0.65)}$ | – | – | – | – |
| ProtoNet+FWT[*](Tseng et al., 2020) | $23.77_{(0.42)}$ | $72.72_{(0.70)}$ | $38.87_{(0.52)}$ | $67.34_{(0.76)}$ | – | – | – | – |
| RelationNet+FWT(BN)[†](Tseng et al., 2020) | $23.95_{(0.20)}$ | $75.78_{(0.40)}$ | $38.68_{(0.30)}$ | $69.13_{(0.40)}$ | $65.55_{(0.40)}$ | $44.29_{(0.30)}$ | $40.18_{(0.40)}$ | $59.77_{(0.40)}$ |
| GNN+FWT[†](Tseng et al., 2020) | $24.28_{(0.20)}$ | $87.07_{(0.40)}$ | $40.87_{(0.40)}$ | $78.02_{(0.40)}$ | $70.70_{(0.50)}$ | $49.66_{(0.40)}$ | $46.19_{(0.40)}$ | $64.97_{(0.50)}$ |
| TPN+FWT(T)[†](Tseng et al., 2020) | $21.22_{(0.10)}$ | $70.06_{(0.70)}$ | $36.96_{(0.40)}$ | $65.69_{(0.50)}$ | $66.75_{(0.50)}$ | $43.20_{(0.50)}$ | $34.03_{(0.40)}$ | $58.18_{(0.50)}$ |
| ATA[†](Wang & Deng, 2021) | $24.43_{(0.20)}$ | $90.59_{(0.30)}$ | $45.83_{(0.30)}$ | $\mathbf{83.75}_{(0.40)}$ | $75.48_{(0.40)}$ | $55.08_{(0.40)}$ | $49.14_{(0.40)}$ | $66.22_{(0.50)}$ |
| LRP-CAN (T)(Sun et al., 2021) | – | – | – | – | $\mathbf{76.90}_{(0.39)}$ | $51.63_{(0.41)}$ | $42.57_{(0.42)}$ | $66.57_{(0.43)}$ |
| LRP-GNN (Sun et al., 2021) | – | – | – | – | $74.45_{(0.47)}$ | $54.46_{(0.46)}$ | $46.20_{(0.46)}$ | $64.44_{(0.48)}$ |
| CHEF(Adler et al., 2020) | $24.72_{(0.14)}$ | $86.87_{(0.20)}$ | $41.26_{(0.34)}$ | $74.15_{(0.27)}$ | – | – | – | – |
| HVM(Du et al., 2022) | $\mathbf{27.15}_{(0.45)}$ | $87.65_{(0.35)}$ | $42.05_{(0.34)}$ | $74.88_{(0.45)}$ | – | – | – | – |
| APPL (T) | $24.87_{(0.41)}$ | $\mathbf{92.51}_{(0.84)}$ | $\mathbf{46.28}_{(0.64)}$ | $79.78_{(0.78)}$ | $68.84_{(0.80)}$ | $\mathbf{55.20}_{(0.58)}$ | $\mathbf{52.67}_{(0.42)}$ | $\mathbf{67.46}_{(0.78)}$ |

epochs with 100 meta-training tasks and 15 query instances per class. Adam optimizer with weight decay of 1e-2 and learning rate of 1e-6 is used to train the APPL. The trade-off parameters $\lambda_{dis}$ and $\lambda_{coh}$ are set to 0.1 and 1e-3 respectively. The proposed APPL is evaluated on 600 randomly selected few-shot learning tasks in each target domain. We fine-tune the feature encoder for 100 iterations for each task with a learning rate of 1e-2. For the fine-tuning hyperparameters, $\alpha_{min}$, $\alpha_0$, $\gamma$ and $\epsilon$ take the values of 0.1, 0.5, 0.99 and 0.4 respectively. For all experiments we report the average accuracy (%) as well as 95% confidence interval.

## 4.2 COMPARISON RESULTS

### 4.2.1 LEARNING WITH FEW SHOTS

We first evaluate the performance of the proposed APPL method on the common cross-domain 5-way 5-shot classification tasks. We compare APPL with both a set of representative FSL methods (MatchingNet (Vinyals et al., 2016), MAML (Finn et al., 2017), ProtoNet (Snell et al., 2017), RelationNet (Sung et al., 2018), MetaOpt (Lee et al., 2019), GNN (Satorras & Estrach, 2018) and TPN (Liu et al., 2019)) and five state-of-the-art CDFSL methods (FWT (Tseng et al., 2020), ATA (Wang & Deng, 2021), LRP (Sun et al., 2021), CHEF (Adler et al., 2020) and HVM (Du et al., 2022)). FWT has been applied jointly with five standard FSL methods: MatchingNet, ProtoNet, RelationNet, GNN and TPN. LRP has been applied jointly with two standard FSL methods: GNN and Cross-attention network (CAN). The comparison results are presented in Table 1, where the top part of the table reports the results of the standard FSL methods and the bottom part reports the results of the CDFSL methods.

We can see that the CDFSL methods (ATA, CHEF, HVM, LRP and APPL) designed specifically to handle vast differences between source and target domains perform largely better than the standard FSL works for in-domain settings. FWT however only produces improvements in most cases over its base RelationNet. Notably, the proposed APPL outperforms all standard FSL methods including ProtoNet and ProtoNet+FWT on all the eight datasets. In particular, its performance gain over ProtoNet is remarkable, exceeding 10% on four out of the eight datasets, which highlights the importance of the prototype learning network. In addition, APPL outperforms all the CDFSL methods on five datasets, and produces the second best results on the two datasets. These results demonstrate the effectiveness of the proposed APPL method for cross-domain few-shot learning.

### 4.2.2 LEARNING WITH HIGHER SHOTS

We further investigated CDFSL with higher-shot tasks in the target domain. In particular, we evaluate the proposed method with cross-domain 5-way 20-shot and 5-way 50-shot learning tasks on four target-domain datasets (ChestX, CropDiseases, ISIC and EuroSAT), on which previous works also reported results. To handle higher-shot problems and increase the scalability of APPL, we extend APPL by adding a clustering function $g$ (details are given in appendix) prior to the PCN

Table 2: Mean classification accuracy (95% confidence interval within brackets) for cross-domain 5-way 20-shot and 50-shot classification. * denotes results reported in Guo et al. (2020). Transductive methods are indicated using (T). Methods sharing query data via Batch Normalization are indicated using (BN)

| | ChestX | | CropDiseases | | ISIC | | EuroSAT | |
|---|---|---|---|---|---|---|---|---|
| | 20-shot | 50-shot | 20-shot | 50-shot | 20-shot | 50-shot | 20-shot | 50-shot |
| MatchingNet*(Vinyals et al., 2016) | $23.61_{(0.86)}$ | $22.12_{(0.88)}$ | $76.38_{(0.67)}$ | $58.53_{(0.73)}$ | $45.72_{(0.53)}$ | $54.58_{(0.65)}$ | $77.10_{(0.57)}$ | $54.44_{(0.67)}$ |
| MAML(BN)*(Finn et al., 2017) | $27.53_{(0.43)}$ | – | $89.75_{(0.42)}$ | – | $52.36_{(0.57)}$ | – | $81.95_{(0.55)}$ | – |
| ProtoNet*(Snell et al., 2017) | $28.21_{(1.15)}$ | $29.32_{(1.12)}$ | $88.15_{(0.51)}$ | $90.81_{(0.43)}$ | $49.50_{(0.55)}$ | $51.99_{(0.52)}$ | $82.27_{(0.57)}$ | $80.48_{(0.57)}$ |
| MetaOpt*(Lee et al., 2019) | $25.53_{(1.02)}$ | $29.35_{(0.99)}$ | $82.89_{(0.54)}$ | $91.76_{(0.38)}$ | $49.42_{(0.60)}$ | $54.80_{(0.54)}$ | $79.19_{(0.62)}$ | $83.62_{(0.58)}$ |
| RelationNet(BN)*(Sung et al., 2018) | $26.63_{(0.92)}$ | $28.45_{(1.20)}$ | $80.45_{(0.64)}$ | $85.08_{(0.53)}$ | $41.77_{(0.49)}$ | $49.32_{(0.51)}$ | $74.43_{(0.66)}$ | $74.91_{(0.58)}$ |
| MatchingNet+FWT*(Tseng et al., 2020) | $23.23_{(0.37)}$ | $23.01_{(0.34)}$ | $74.90_{(0.71)}$ | $75.68_{(0.78)}$ | $32.01_{(0.48)}$ | $33.17_{(0.43)}$ | $63.38_{(0.69)}$ | $62.75_{(0.76)}$ |
| ProtoNet+FWT*(Tseng et al., 2020) | $26.87_{(0.43)}$ | $30.12_{(0.46)}$ | $85.82_{(0.51)}$ | $87.17_{(0.50)}$ | $43.78_{(0.47)}$ | $49.84_{(0.51)}$ | $75.74_{(0.70)}$ | $78.64_{(0.57)}$ |
| RelationNet+FWT(BN)*(Tseng et al., 2020) | $26.75_{(0.41)}$ | $27.56_{(0.40)}$ | $78.43_{(0.59)}$ | $81.14_{(0.56)}$ | $43.31_{(0.51)}$ | $46.38_{(0.53)}$ | $69.40_{(0.64)}$ | $73.84_{(0.60)}$ |
| CHEF(Adler et al., 2020) | $29.71_{(0.27)}$ | $31.25_{(0.20)}$ | $94.78_{(0.12)}$ | $96.77_{(0.08)}$ | $54.30_{(0.34)}$ | $60.86_{(0.18)}$ | $83.31_{(0.14)}$ | $86.55_{(0.15)}$ |
| HVM(Du et al., 2022) | $30.54_{(0.47)}$ | $32.76_{(0.46)}$ | $95.13_{(0.35)}$ | $97.83_{(0.33)}$ | $54.97_{(0.35)}$ | $61.71_{(0.32)}$ | $84.81_{(0.34)}$ | $87.16_{(0.35)}$ |
| APPL(T) | $\mathbf{30.75}_{(0.41)}$ | $\mathbf{33.14}_{(0.88)}$ | $\mathbf{95.77}_{(0.53)}$ | $\mathbf{98.14}_{(0.56)}$ | $\mathbf{57.97}_{(0.73)}$ | $\mathbf{62.17}_{(0.43)}$ | $\mathbf{88.60}_{(0.85)}$ | $\mathbf{89.75}_{(0.76)}$ |

component. The $g$ function clusters the support instances in each class into $K' = 5$ clusters based on their learned embeddings. The obtained cluster centroid vectors are then concatenated as input for PCN. We compared APPL with both standard FSL methods and several CDFSL methods, and the results are presented in Table 2, where the top part of the table presents the results of the FSL methods and the bottom part presents the results of the CDFSL methods.

We can see that again the CDFSL methods (CHEF, HVM, and APPL) outperform the standard FSL methods. The performance gains of APPL over Protonet and Protonet+FWT are remarkable exceeding 6% and 9% on three datasets (CropDiseases, ISIC and EuroSAT) in the cases of 20-shot and 50-shot respectively. Moreover, APPL consistently outperforms all the other methods on all the four datasets for both the 20-shot and 50-shot cases. These results again validate the effectiveness of APPL for cross-domain few-shot learning and demonstrate its capacity in handling cross-domain higher-shot learning problems. Two factors account for our proposed method's good performance in the case of higher shots: First, benefiting from the proposed WMA self-training approach in the target domain, we are able to generate more accurate pseudo-labels with higher shots, which enables our model to obtain better results. Second, we conduct clustering over the embeddings of support instances to generate class centroids with higher shots, which can eliminate some noisy information and allow the PCN to learn the most representative features.

### 4.3 ABLATION STUDY

To investigate the importance of each component of the proposed APPL approach, we conducted an ablation study to compare APPL with its six variants: (1) "$-$w/o $\psi_\phi$", which drops PCN and replaces it with a simple average of the support instances of each class. (2) "$-$w/o $\mathcal{L}_{dis}$" and (3) "$-$w/o $\mathcal{L}_{coh}$", which drop the $\mathcal{L}_{dis}$ loss and $\mathcal{L}_{coh}$ loss respectively. (4) "$-$w/o $\mathcal{L}_{CE}(S)$", which drops the cross-entropy loss over the support instances in fine-tuning. (5) "$-$w/o $\mathcal{L}^{tr}_{CE}(Q)$", which drops the cross-entropy loss over the query instances and hence the WMA self-training component in fine-tuning. (6) "ProtoNet", which can be considered as a variant of APPL that drops both PCN and WMA self-training, as well as $\mathcal{L}_{dis}$ and $\mathcal{L}_{coh}$.

We compared APPL with its six variants on the cross-domain 5-way 5-shot setting on all the eight datasets, and the results are reported in Table 3. We can see that APPL outperforms all the other variants on almost all datasets. The "$-$w/o $\mathcal{L}_{CE}(S)$" variant produced the largest performance drop among all variants, which highlights the importance of the few labeled support instances for fine-tuning in the target domain. The performance degradation for ProtoNet and "$-$w/o $\psi_\phi$" highlights the importance of the proposed PCN component. In addition, "$-$w/o $\psi_\phi$" outperforms ProtoNet, which underlines the performance gain obtained by using pseudo-labeled query instances with WMA self-training and the prototype regularization losses in the absence of PCN. The other three variants, "$-$w/o $\mathcal{L}_{dis}$", "$-$w/o $\mathcal{L}_{coh}$" and "$-$w/o $\mathcal{L}^{tr}_{CE}(Q)$", also perform worse than APPL, which verifies the contributions of the two prototype regularization loss terms and the WMA self-training component respectively.

Table 3: Ablation study results for cross-domain 5-way 5-shot classification tasks.

| | ChestX | CropDisea. | ISIC | EuroSAT | Places | Planate | Cars | CUB |
|---|---|---|---|---|---|---|---|---|
| ProtoNet | $24.05_{(1.01)}$ | $79.72_{(0.67)}$ | $39.57_{(0.57)}$ | $73.29_{(0.71)}$ | $58.54_{(0.68)}$ | $46.80_{(0.65)}$ | $41.74_{(0.72)}$ | $55.51_{(0.68)}$ |
| APPL | $\mathbf{24.87}_{(0.41)}$ | $\mathbf{92.51}_{(0.84)}$ | $\mathbf{46.28}_{(0.64)}$ | $\mathbf{79.78}_{(0.78)}$ | $\mathbf{68.84}_{(0.80)}$ | $55.20_{(0.58)}$ | $\mathbf{51.67}_{(0.42)}$ | $\mathbf{67.46}_{(0.78)}$ |
| $-$w/o $\psi_\phi$ | $22.33_{(0.56)}$ | $89.11_{(0.66)}$ | $43.99_{(0.68)}$ | $77.99_{(0.68)}$ | $67.57_{(0.33)}$ | $53.69_{(0.60)}$ | $50.30_{(0.81)}$ | $64.03_{(0.97)}$ |
| $-$w/o $\mathcal{L}_{dis}$ | $24.84_{(0.69)}$ | $91.31_{(0.72)}$ | $43.23_{(0.82)}$ | $78.25_{(0.76)}$ | $66.18_{(0.49)}$ | $54.56_{(0.84)}$ | $51.56_{(0.85)}$ | $60.45_{(0.85)}$ |
| $-$w/o $\mathcal{L}_{coh}$ | $23.77_{(0.68)}$ | $90.19_{(0.76)}$ | $43.69_{(0.49)}$ | $79.53_{(0.71)}$ | $65.89_{(0.47)}$ | $51.62_{(0.77)}$ | $50.32_{(0.81)}$ | $60.96_{(0.85)}$ |
| $-$w/o $\mathcal{L}_{CE}(S)$ | $21.08_{(0.42)}$ | $59.15_{(0.59)}$ | $26.90_{(0.78)}$ | $49.95_{(0.80)}$ | $34.79_{(0.85)}$ | $25.72_{(0.77)}$ | $27.19_{(0.87)}$ | $30.12_{(0.74)}$ |
| $-$w/o $\mathcal{L}_{CE}^{tr}(Q)$ | $22.36_{(0.42)}$ | $88.24_{(0.86)}$ | $42.14_{(0.59)}$ | $77.63_{(0.65)}$ | $67.14_{(0.82)}$ | $\mathbf{55.69}_{(0.60)}$ | $51.59_{(0.82)}$ | $60.22_{(0.85)}$ |

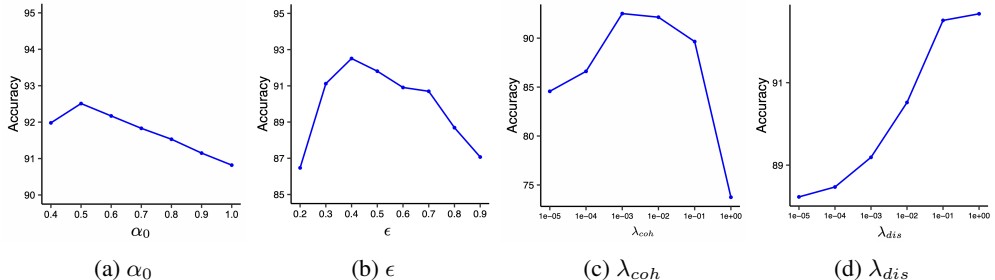

(a) $\alpha_0$      (b) $\epsilon$      (c) $\lambda_{coh}$      (d) $\lambda_{dis}$

Figure 2: Sensitivity analysis for the proposed method on hyper-parameters $\alpha_0$, $\epsilon$, $\lambda_{coh}$ and $\lambda_{dis}$ on CropDisease dataset under cross-domain 5-way 5-shot task: (a) $\alpha_0$, (b) $\epsilon$, (c) $\lambda_{coh}$, (d) $\lambda_{dis}$.

## 5 HYPER-PARAMETER SENSITIVITY ANALYSIS

To demonstrate the effect of the hyper-parameters of our proposed method, we summarize the results of various studies in Figure 2. The figure shows the performance of our proposed method on CropDisease dataset under the cross-domain 5-way 5-shot task as we modify each hyper-parameter separately. As seen from the results, it is clear that our proposed APPL is not sensitive to the choice of value for $\alpha_0$, as the performance of APPL is stable across all values of $\alpha_0$. As for $\lambda_{coh}$, too large or too small a value will pull the query/support instances to the prototypes more strongly or more weakly, thus having a negative impact on the results. A reasonable value between 1e-1 and 5e-3 is required to obtain reasonable performance. In the case of $\lambda_{dis}$, the experimental results improve as the value of $\lambda_{dis}$ increases. And the results become stable when the parameter reaches 1e-1. We believe that this is because $\lambda_{dis}$ controls the scale of pushing the prototypes of different classes away from each other so that when the distance between the prototypes reach a certain threshold, embedding is no longer susceptible to mutual influence.

Finally, it is worth noting that $\epsilon$ is an important hyperparameter. $\epsilon$ represents the degree of certainty of the pseudo-labels' predictions utilized in fine-tuning. When $\epsilon$ is too small, more unlabeled samples are used in fine-tuning with their noisy pseudo-labels, and when $\epsilon$ is too large, less unlabeled samples are used in fine-tuning with less noisy pseudo-labels. Experimental results show that when the intermediate value of 0.4 is selected, the number and accuracy of pseudo-labels can be balanced and the best results for this model were obtained.

## 6 CONCLUSION

In this paper, we proposed a novel Adaptive Parametric Prototype Learning (APPL) method to address the cross-domain few-shot learning problem. APPL meta-trains an adaptive prototype calculator network in the source domain to learn more discriminative and representative class prototypes from various support instance layouts, which can then guide the feature encoder to adapt to the target domain through fine-tuning. Moreover, a weighted-moving-average self-training approach is adopted to enhance fine-tuning by exploiting the unlabeled query instances in the target domain to mitigate domain shift and avoid overfitting to support instances. Experimental results on eight benchmark cross-domain few-shot classification datasets demonstrate that the proposed APPL outperforms existing state-of-the-art methods.

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

---

**Algorithm 2** Training Procedure on Source Domain

---

**Input**: Source domain dataset $D_s$, $K$, $N$; pre-trained feature extractor $f_\theta$;
      learning-rate $\gamma_1$ and $\gamma_2$;
**Output**: Learned model parameters $\theta, \phi$
**Initialize**: $\phi \leftarrow \phi_0, \theta \leftarrow \theta_0$
**for** iter = 1 **to** maxiters **do**
    $V \leftarrow$ randomly sample N class indices from all classes in $D_s$
    **for** $n$ in $\{1, .., N\}$ **do**
        $S_n, Q_n \leftarrow$ randomly sample support & query sets for class $n$ from $D_s^V$
    **end for**
    $S = S_1 \cup .. \cup S_N, \quad Q = Q_1 \cup .. \cup Q_N$
    **for** initer=1 **to** maxiniters **do**
        $\theta \leftarrow \theta - \gamma_1 \nabla_\theta \mathcal{L}_{CE}(S)$
    **end for**
    Compute $\mathcal{L}_{dis}$ and $\mathcal{L}_{coh}$ with Eq. (4) and Eq.(5) respectively.
    $\mathcal{L}_{train} = \lambda_{dis}\mathcal{L}_{dis} + \lambda_{coh}\mathcal{L}_{dis}$
    **for** $(x, y) \in Q$ **do**
        $\mathcal{L}_{train} \leftarrow \mathcal{L}_{train} + \ell_{CE}(x, y)$
    **end for**
    $\phi \leftarrow \phi - \gamma_2 \nabla_\phi \mathcal{L}_{train}$
**end for**

---

## A    META-TRAINING ON SOURCE DOMAIN

We present the meta-training procedure for our proposed APPL in Algorithm 2.

## B    DETAILS ON LEARNING WITH HIGHER SHOTS

In order to demonstrate the scalability of the proposed APPL in the case of higher shots experimental setup, we propose a clustering-based solution to guarantee that the Adaptive Prototype Calculator Network scales efficiently with the number of support samples.

Specifically, we first cluster the embeddings of support instances of each class into $K'$ clusters as follows:

$$\mu = g(f_\theta(x_1^i), .., f_\theta(x_K^i)), \tag{14}$$

where $g$ is the clustering function that takes as input the learned embeddings of the support instances of a given class. As a result, the class centroids $\mu = (\mu_1, \mu_2, .., \mu_{K'})$ are obtained, where $K'$ is the number of clusters. In this process, the clustering function $g$ learns the cluster centroids by minimizing the following loss function:

$$\min_{\mu,c} \mathcal{L}_{clust} = \sum_{j=1}^{K} \sum_{\ell=1}^{K'} \mathbb{1}_{(c_j=\ell)} ||f_\theta(x_j) - \mu_\ell||^2, \tag{15}$$

where $c$ is the clustering assignment vector. Then, we use the concatenation of the obtained cluster centroid vectors as input to the adaptive prototype calculator network as follows:

$$p_i = \psi_\phi(conct(\mu_1, \mu_2, .., \mu_{K'})), \tag{16}$$

Consequently, the number of learnable parameters of our adaptive prototype calculator network is fixed regardless of the number of support instances in each class, which ensures the proposed APPL to be scalable and easy to apply in the cases of higher shots.

## C    IMPACT OF THE INPUT ORDER OF ADAPTIVE PROTOTYPE CALCULATOR NETWORK

To demonstrate the effectiveness of the order of the input samples fed to the adaptive prototype calculator network, we summarize the results in Table 4. Since the concatenated embeddings of the

support set instances are fed as input to the adaptive prototype calculator network to generate the prototypes of each class, it is important to demonstrate that APPL is resilient to the concatenation order of the support samples. Therefore, we evaluate the performance of APPL under the cross-domain 5-way 5-shot learning task under 4 different datasets where the order of the input samples to the adaptive prototype calculator network is randomly permuted during the evaluation step of the fine-tuning step. We generate 5 different random permutations of the input samples and report the corresponding results for each permutation in Table 4.

Table 4: Input order of Adaptive Prototype Calculator Network results in terms of mean classification accuracy (95% confidence interval within brackets) on 4 target domain datasets using the cross-domain 5-way 5-shot task.

|  | ChestX | CropDiseases | ISIC | EuroSAT |
|---|---|---|---|---|
| Permutation # 1 | $24.79_{(0.36)}$ | $92.37_{(0.77)}$ | $46.12_{(0.55)}$ | $79.69_{(0.72)}$ |
| Permutation # 2 | $24.68_{(0.35)}$ | $92.73_{(0.78)}$ | $46.29_{(0.54)}$ | $79.57_{(0.72)}$ |
| Permutation # 3 | $24.82_{(0.36)}$ | $92.52_{(0.77)}$ | $46.28_{(0.56)}$ | $79.68_{(0.74)}$ |
| Permutation # 4 | $24.90_{(0.36)}$ | $92.54_{(0.76)}$ | $46.18_{(0.55)}$ | $79.79_{(0.74)}$ |
| Permutation # 5 | $24.87_{(0.41)}$ | $92.51_{(0.84)}$ | $46.28_{(0.64)}$ | $79.78_{(0.78)}$ |

