# OpenReview forum: "Adaptive Parametric Prototype Learning for Cross-Domain Few-Shot Classification"
_ICLR.cc/2023/Conference — Submitted to ICLR 2023_

### Official Review · Reviewer_9xQ2 · 2022-10-16

**Confidence:** 5
**Correctness:** 3
**Technical Novelty And Significance:** 2
**Empirical Novelty And Significance:** 2
**Recommendation:** 3

**Clarity, Quality, Novelty And Reproducibility:**

The overall novelty and quality of this paper do not surpass the ICLR acceptance bar.

Technical contributions are insufficient, and experimental improvements are also marginal (detailed above).



**Strength And Weaknesses:**

Strength:
- This paper proposed the Adaptive Parametric Prototype Learning (APPL) method, which includes PCN and WMA submodules designed for the CDFSL problem.
- Experiments are conducted on eight datasets, and slightly improved results are obtained on some datasets.

Weaknesses:
- Technically, the proposed modules (PCN and WMA) provide little new knowledge for the few-shot and cross-domain few-shot learning field. A series of improvements for Prototypical Networks is proposed, e.g., [R1-R3] to name a few. Learning from the concatenated support features is similar to set-to-set representation [R4]. WMA explored the idea of transduction with moving average updates, which was also discovered in previous work [R5, R6].
- Experimentally, the performance improvements are marginal or negative on most dataets, e.g., in Table 1, the proposed method underperforms baselines on three datasets, overlaps on three datasets, and only outperforms with a significant margin on two datasets. Similar results are in Table 2.
- This paper builds on the cross-domain setting of Guo et al. (2020). However, as a cross-domain few-shot learning paper, it should also discuss other cross-domain settings, at least in the related work section, e.g., [R2, R7-R10].
- In Table 1 and Table 2, it is better to indicate the transductive/inductive setting of each method.
- Typos. In Sec 2.2., (1) "... methods. for cross-domain few-shot learning . The FWT ...", (2) "... across largely different domains .", (3) in Figure 1, "(a)" and "b)" are inconsistent.





[R1] Yang, Boyu, et al. "Prototype mixture models for few-shot semantic segmentation." European Conference on Computer Vision. Springer, Cham, 2020.
[R2] Bateni, Peyman, et al. "Improved few-shot visual classification." Proceedings of the IEEE/CVF Conference on Computer Vision and Pattern Recognition. 2020.
[R3] Jiang, Ruijin, and Zhaohui Cheng. "Mixture Gaussian prototypes for few-shot learning." 2021 International Conference on Data Mining Workshops (ICDMW). IEEE, 2021.
[R4] Ye, Han-Jia, et al. "Few-shot learning via embedding adaptation with set-to-set functions." Proceedings of the IEEE/CVF Conference on Computer Vision and Pattern Recognition. 2020.
[R5] Chen, Wentao, et al. "Few-Shot Learning with Part Discovery and Augmentation from Unlabeled Images." arXiv preprint arXiv:2105.11874 (2021).
[R6] Islam, Ashraful, et al. "Dynamic distillation network for cross-domain few-shot recognition with unlabeled data." Advances in Neural Information Processing Systems 34 (2021): 3584-3595.
[R7] Triantafillou, Eleni, et al. "Meta-dataset: A dataset of datasets for learning to learn from few examples." arXiv preprint arXiv:1903.03096 (2019).
[R8] Liu, Yanbin, et al. "A multi-mode modulator for multi-domain few-shot classification." Proceedings of the IEEE/CVF International Conference on Computer Vision. 2021.
[R9] Bateni, Peyman, et al. "Improving few-shot visual classification with unlabelled examples." (2020).
[R10] Doersch, Carl, Ankush Gupta, and Andrew Zisserman. "Crosstransformers: spatially-aware few-shot transfer." Advances in Neural Information Processing Systems 33 (2020): 21981-21993.


**Summary Of The Paper:**

This paper proposes an adaptive parametric prototype learning approach for one specific setting of the cross-domain few-shot learning (CDFSL) classification.

Technical contributions: (1) the authors proposed the prototype calculator network (PCN) to learn new prototypes for the support set with meta-training; (2) a weighted-moving-average (WMA) self-training strategy was proposed and applied to the query images at meta-test time in a transductive manner.

Experiments: the proposed algorithm is evaluated on both Guo et al. (2020) and FWT (Tseng et al., 2020) settings for CDFSL. Higher performances are achieved on the part of the eight investigated datasets with different improvements. An ablation study and parameter analysis are conducted.

**Summary Of The Review:**

The technical contributions are insufficient for an ICLR paper, the experimental verification is also less evident.

I think this paper is not ready for publishing in ICLR.

---

> ### Author Response · Authors · 2022-11-18
> **Response to Reviewer 9xQ2 (Part 1)**
>
> * **Technically, the proposed modules (PCN and WMA) provide little new knowledge for the few-shot and cross-domain few-shot learning field. A series of improvements for Prototypical Networks is proposed, e.g., [R1-R3] to name a few. Learning from the concatenated support features is similar to set-to-set representation [R4]. WMA explored the idea of transduction with moving average updates, which was also discovered in previous work [R5, R6].**
>
> We start by stating **the novelty** of the paper, which presents a new APPL method for the CDFSL problem. APPL meta-trains a novel adaptive prototype calculator network (PCN) to learn more discriminative and representative class prototypes, and devises a new weighted-moving-average self-training approach to mitigate domain shift and avoid overfitting by exploiting the unlabeled query instances. Our method is very different from existing methods and offers a new perspective to achieve state-of-the-art performance for cross domain few-shot learning.
>
> For the prototype network, existing methods [R1-R3] focus on generating prototypes from the perspective of a mixed Gaussian model [R1,R3] or different metric distances [R2] on the same domain or similar domains. However, they lack consideration of the correlation of prototypes among the same and different categories, and their proposed approaches have difficulty handling large cross-domain shifts and adapting to target domains. **Instead**, in our proposed PCN, we use a **parameterization mechanism** to generate more representative prototypes and propose two loss functions to learn discriminative class prototypes by **enforcing both inter-class discriminability and intra-class cohesion in the extracted feature space.**
>
> Regarding concatenated support features, Our model is very different from FEAT [R4] in both **motivation and implementation**. In terms of motivation, FEAT’s motivation is to learn task-specific embeddings of all the support instances using set-to-set functions. **In contrast**, our method exploits **learning discriminative class prototypes** by enforcing both inter-class discriminability and intra-class cohesion. As for implementation, FEAT applies their set-to-set functions jointly to the entire support set by employing Transformer, a computationally expensive model. Whereas our method employs **simple neural network architecture** to implement our Prototype Calculator Network. Furthermore, FEAT learns task-adaptive embeddings which are close to their class center therefore only taking in consideration intra-class cohesion, while our Prototype Calculator Network learns class prototypes by enforcing **both inter-class discriminability and intra-class cohesion.**
>
> Regarding the moving-average updates, our proposed WMA self-training strategy is **specifically designed**, rather than applied directly from existing methods. Existing methods demand lots of additional data to implement self-training. For example, both [R5] and [R6] require collecting additional unlabeled samples to train their proposed models, from a base dataset in [R5] and from target domain in [R6]. Such requirements are hard to meet and hence hamper their applicability in many domains. **In contrast**, our WMA self-training strategy uses **only available query data**. Besides, [R5] and [R6] apply moving-average updates (momentum) to their model parameters while we apply weighted-moving-average to **calculate robust/stable pseudo-labels**. **Our vital contribution** is that adopting the weighted moving average directly on the predictions of the model is a novel update mechanism for generating pseudo-labels, which can stabilize the self-training process and dampen possible oscillating predictions for challenging query instances. Finally, our WMA self-training is equipped with a **rectified annealing** schedule to adjust the momentum hyper-parameter to stabilize the produced pseudo-labels, while the moving average updates to the model parameters in the two aforementioned works do not adopt a similar strategy.
>
> Consequently, our proposed APPL method is novel and has better generalization capacity in learning class prototypes in the feature embedding space, and hence effectively mitigates the domain shift and adapts to the target domain with WMA self-training.

---

> > ### Author Response · Authors · 2022-11-18
> > **Response to Reviewer 9xQ2 (Part2)**
> >
> > * **Experimentally, the performance improvements are marginal or negative on most dataets, e.g., in Table 1, the proposed method underperforms baselines on three datasets, overlaps on three datasets, and only outperforms with a significant margin on two datasets. Similar results are in Table 2.**
> >
> > We argue that our method achieves the best results on five datasets and the second-best results on two datasets in Table1. In addition, our method achieves the best results on all datasets in Table2. Even though the 5-shot results of ATA (Wang & Deng, 2021) are better than ours on EuroSAT and Places datasets, ATA is built on top of GNN or TPN. These two baselines already outperform many methods on these two datasets, so it would be inappropriate to say that we are worse than ATA. Besides, ATA is applied jointly with 2 other CDFSL methods (FWT and LRP) in order to obtain its good performance. In contrast, our proposed method is standalone and does not require to be applied jointly with any other CDFSL methods. Furthermore, ATA is computationally expensive where its number of parameters scales linearly with the number of shots. Instead, our proposed method scales efficiently to learning with higher shots where its number of parameters is independent of the number of shots. In the case of learning with higher shots, APPL consistently outperforms all other methods in both the 20-shot and 50-shot scenarios on all 4 datasets. Consequently,our proposed APPL method is effective and has scalability and superior performance for addressing the cross-domain few-shot learning problem.
> >
> > * **This paper builds on the cross-domain setting of Guo et al. (2020). However, as a cross-domain few-shot learning paper, it should also discuss other cross-domain settings, at least in the related work section, e.g., [R2, R7-R10].**
> >
> > We have added the following to the Cross-Domain Few-shot Learning subsection of Related Works section:
> >
> > Some other works [R2,R7-R10] have tested alternative cross-domain few-shot learning settings such as Meta-Dataset [R7] CDFSL setting where models are trained on several source-domain datasets and tested on multiple target-domain datasets. In this work, we focus on the CDFSL setting in Guo et al (2020) as it is the most widely studied CDFSL setting.
> >
> > * **In Table 1 and Table 2, it is better to indicate the transductive/inductive setting of each method.**
> >
> > Based on your suggestions, we have indicated transductive methods in Table 1 and Table 2.
> >
> > * **Typos. In Sec 2.2., (1) "... methods. for cross-domain few-shot learning . The FWT ...", (2) "... across largely different domains .", (3) in Figure 1, "(a)" and "b)" are inconsistent**
> >
> > Thank you for pointing out the typos, we fixed the typos you mentioned.

---

### Official Review · Reviewer_BfM8 · 2022-10-23

**Confidence:** 4
**Correctness:** 3
**Technical Novelty And Significance:** 2
**Empirical Novelty And Significance:** 2
**Recommendation:** 5

**Clarity, Quality, Novelty And Reproducibility:**

Clarity of this work is not very good. Motivations and contributions are not well discussed. In the section of methods, statements, formulations and algorithms are not well organized.
Quality of this work is at borderline level and novelty of this paper is a bit limited as discussed above.
Reproducibility: hope the author can release code of the proposed method for reproducibility if accepted.


**Strength And Weaknesses:**

Strength:
+ The studied cross-domain few-shot problem is more challenge than simple few-shot classification problem. Methods for CDFS are more robust towards real scenarios.
+ Experiments are conducted on eight target domain datasets. It’s more complete than some previous works. And performance of the proposed method is competitive.

Weaknesses:
- The proposed method combines some losses and self-training method which is widely used in domain adaptation with prototypical networks. Novelty of this work is a bit limited.
- Authors didn’t summarize their contributions and the superiority of this work over previous few-shot and cross-domain few-shot methods, since this paper is not the first to tackle cross-domain few-shot classification problem.
- There are too many losses and hyperparameters in the proposed method, searching for the best combination of hyperparameters may be hard.
- A few mistakes in writing. E.g. λ in Fig.1. or L ? Quotation marks in section 4.3 are incorrect. Chinese quotation marks are used by the authors.


**Summary Of The Paper:**

This paper tackles the problem of cross-domain few-shot classification, in which novel classes and base classes belong to different data distribution. The authors propose a novel Adaptive Parametric Prototype Learning method based on prototypical network. For meta training, class prototypes are learned by enforcing both inter-class discriminability and intra-class cohesion. For finetuning approach, weight-moving-average self-training is applied. Experiments are conducted on a single source dataset and eight target datasets. Performance of the proposed method is competitive against other few-shot or cross-domain few-shot learning methods.

**Summary Of The Review:**

This paper studies a challenge problem of cross-domain few-shot classification. However, the problem is studied by many previous works. Methods proposed by this work can achieve better performance than most of previous works, but the contribution and novelty towards CDFS problem is limited. In addition, this work has some problem with writing and clarity. So from my perspective, it is below the acceptance threshold.

---

> ### Author Response · Authors · 2022-11-18
> **Response to Reviewer BfM8**
>
> Thank you for acknowledging the importance of problem of cross-domain few-shot learning and robustness of our approach
> * **The proposed method combines some losses and self-training method which is widely used in domain adaptation with prototypical networks. Novelty of this work is a bit limited.**
>
> We would like to emphasize that our proposed loss function is specifically designed, rather than chosen and applied directly from existing methods to the CDFSL problem. First, they are computed over the prototypes learned via the parametric mechanism of different classes, which are the global representation for the classes based on the concatenated feature vectors of the support instances. Second, instead of just being able to be used in the source domain, they also work perfectly with the pseudo labels that we generate in the target domain to help in the fine-tuning process of the model. As for the proposed self-training method, existing methods require collecting a large amount of additional data to implement self-training. In contrast, our WMA self-training strategy uses only available query data. Additionally, our proposed self-training method ensures that the produced pseudo-labels are stable and clean (not noisy) by jointly employing three mechanisms: weighted-moving-average updating of prediction vectors, a rectified annealing schedule for the WMA and selectively sampling only the confident pseudo-labels to adapt the model.
>
> * **Authors didn’t summarize their contributions and the superiority of this work over previous few-shot and cross-domain few-shot methods, since this paper is not the first to tackle cross-domain few-shot classification problem.**
>
>  Although we are not the first one to address the CDFSL problem, our proposed method is very different from existing methods and overcomes some of the drawbacks that still exist in existing methods. The contributions of our proposed method are as follows:
> 1.  We propose a novel adaptive prototype calculator network called Prototype Calculator Network (PCN). Our key contribution is that we use a parameterization mechanism to generate more representative prototypes and propose two loss functions to learn discriminative class prototypes by enforcing both inter-class discriminability and intra-class cohesion in the extracted feature space.
> 2. We propose a WMA self-training strategy that is tailored for the CDFSL problem. Compared to existing methods, it overcomes the barrier of requiring large amounts of additional data for the target domain and reduces domain shift by generating better pseudo-labels. It ensures that the produced pseudo-labels are stable and clean (not noisy) by jointly employing three mechanisms: weighted-moving-average updating of prediction vectors, a rectified annealing schedule for the WMA and selectively sampling only the confident pseudo-labels to adapt the model.
> 3. Our proposed method outperforms existing works on both few-shot (5-shot) and high-shot (20-shot and 50-shot) classification tasks.
>
>
> * **There are too many losses and hyperparameters in the proposed method, searching for the best combination of hyperparameters may be hard.**
>
> There are actually only four hyper-parameters to be considered, and most of them have a very limited range, so this does not make it very difficult to tune the hyper-parameters. More importantly, we conducted hyper-parameter sensitivity analysis, presented in Section 5, where the results demonstrate that our proposed method is robust to the hyper-parameter values achieving good performance across a reasonably wide range of hyper-parameter values.
>
> * **A few mistakes in writing. E.g. λ in Fig.1. or L ? Quotation marks in section 4.3 are incorrect. Chinese quotation marks are used by the authors.**
>
> Thank you for pointing out the typos, we fixed the typos you mentioned
>
>
> * **Clarity of this work is not very good. Motivations and contributions are not well discussed
> In the section of methods, statements, formulations and algorithms are not well organized.**
>
>  Thank you for your valuable comments. We have made the contribution part of our paper clearer based on your valuable suggestions.

---

> > ### Comment · Reviewer_BfM8 · 2022-11-24
> > **Reply to authors**
> >
> > Thank you for your responses. The responses have partly addressed my concerns of novelty of this work. Although methods are old, they are designed specially for the CDFSL problem. However, few new knowledge is brought to the field of CDFSL by this work. And the clarity problem of motivations of many design choices is still a consideration. Overall, I will keep my borderline score of this work.

---

### Official Review · Reviewer_EF7b · 2022-10-23

**Confidence:** 4
**Correctness:** 3
**Technical Novelty And Significance:** 3
**Empirical Novelty And Significance:** 3
**Recommendation:** 5

**Clarity, Quality, Novelty And Reproducibility:**

The writing of the paper is fair and the organization needs improvement, especially section 3.2.2. The quality of the paper is OK. The idea is somewhat novel but the introduction of many hyperparameters makes the method not applicable in real-world applications. The paper seems not easy to reproduce.

**Strength And Weaknesses:**

Strength:
1. The problem of cross-domain few-shot learning is important and the authors propose to improve ProtoNet for cross-domain few-shot learning.
2.  The authors conducted extensive experiments to show the benefits of the proposed approach.


Weaknesses:
1. Many design choices lack motivation. For example,
i. What's the benefits of using the concatenated feature vectors for computing the prototypes? What's the issue of using the feature from the last layer?
ii. Why use the unlabeled target can reduce the domain shift?
iii. What's the motivation of using a weighted-moving-avarage?

2. The proposed method has many hyperparameters which it hard to use the method for other applications.

3. What is the time complexity of the proposed approach compared with the baselines? The introduction of the adaptive prototype calculator network is an additional overhead.


**Summary Of The Paper:**

In this paper, the authors propose a novel Adaptive Parametric Prototype Learning for cross-domain few-shot learning. The idea is to learn class prototypes from a parametric prototype calculator network. The class prototypes  are learnt from concatenated feature vectors of the support instances to ensure inter-class discriminability and intra-class cohesion. The proposed approach is shown to be effective for cross-domain few-shot learning.

**Summary Of The Review:**

 The authors propose a novel Adaptive Parametric Prototype Learning for cross-domain few-shot learning which has some novel ideas. However, the motivations are not clearly stated and it is hard to understand where is the improvement comes from.

---

> ### Author Response · Authors · 2022-11-18
> **Response to Reviewer EF7b**
>
> Thank you for acknowledging the importance of the problem of cross-domain few-shot learning and the novelty of our approach.
>
>
> *  **Many design choices lack motivation. For example, i. What's the benefits of using the concatenated feature vectors for computing the prototypes? What's the issue of using the feature from the last layer? ii. Why use the unlabeled target can reduce the domain shift? iii. What's the motivation of using a weighted-moving-avarage?**
>
> (i)The benefits of using the concatenated feature vectors for computing the prototypes are as follows. First, the concatenation of the features from the last layer of all support samples can ensure our PCN takes into account all the embeddings of the support samples of a given class when calculating the corresponding class prototype. Second, some existing methods simply average the support instances to calculate the class prototypes, which has the evident drawback of ignoring the inter-class and intra-class instance relations. In contrast, our proposed method can learn class prototypes from the support instances through a parametric prototype calculator network (PCN) by enforcing inter-class discriminability and intra-class cohesion from the feature space, producing more representative class prototypes from various support instance layouts.
> Instead of concatenating the features of different layers together as features of one support sample, we concatenate the features of the last layer of all the support samples and input them to the PCN. Therefore, there is no issue with using the features from the last layer.
>
> (ii)There are significant distribution discrepancies between the source and target domains in the CDFSL problem. We employ the unlabeled query instances with predicted soft pseudo-labels to increase the size and diversity of the target data available for fine-tuning and model adaptation, hence mitigate the domain shift between the source and target domains. The Experimental results show that the unlabelled targets do improve the results of the model.
>
> (iii)The motivation behind WMA Self-training: For CDFSL problem, significant distribution discrepancies in the input image space typically exist between the source and target domains. Even worse, since the ground-truth is undefined for unlabeled examples in the target domain, the noise regularization by itself does not help self-training. In this case, the information obtained by the model by considering only the current iteration is not accurate. If it only considers the current generated target, the model suffers from confirmation bias problems as it receives a misleading guide of inconsistent predictions and thus encounters a barrier of difficulty in learning new information. Our WMA self-training strategy can be thought of as an ensemble of the present version of the model and the previous versions that evaluated the same example, which can improve the quality of the predictions. As a result, it can stabilize the self-training process and dampen possible oscillating predictions for challenging query instances.
>
>
> * **The proposed method has many hyperparameters which it hard to use the method for other applications.**
>
> Our proposed method has 4 hyper-parameters, which is not that many for a deep learning model. More importantly, we conducted hyper-parameter sensitivity analysis, presented in Section 5, where the results demonstrate that our proposed method is robust to the hyper-parameter values achieving good performance across a reasonably wide range of hyper-parameter values. Consequently, we think it's not hard to use our approach on other applications.
>
> * **What is the time complexity of the proposed approach compared with the baselines? The introduction of the adaptive prototype calculator network is an additional overhead.**
>
> Our proposed Prototype Calculator Network (PCN) is a simple neural network made up of a single linear layer followed by ReLU activation. Therefore, the number of parameters of PCN is equal to $\mathcal{O}(K.D^2)$ where K is the number of shots and D is the size of the learned embedding by the feature encoder. To ensure the scalability of our proposed PCN to learning with higher shots, clustering function $g$ is employed which reduces the number of parameters of PCN to $\mathcal{O}(K’*D^2)$ where $K’$ is the number of clusters learned by $g$. In our implementation, we use $K’=5$ for both 20-shot and 50-shot learning tasks. Therefore, our proposed method is simple, scalable and efficient. Furthermore, we calculate the number of parameters and FLOPS of our proposed method APPL and compare it with those of the ProtoNet baseline and report the results in the following table. The table clearly shows that our proposed method does not cause a significant increase in FLOPS or # of parameters.
>
>
> |           |   FLOPS  | # of Parameters |
> |-----------|:--------:|:---------------:|
> | ProtoNet  | 0.8956G  |      4.906M     |
> | APPL      | 0.8969G  |      6.217M     |

---

### Official Review · Reviewer_TtNB · 2022-10-28

**Confidence:** 4
**Clarity, Quality, Novelty And Reproducibility:** The approach seems fairly clear and o…
**Correctness:** 4
**Technical Novelty And Significance:** 3
**Empirical Novelty And Significance:** 3
**Recommendation:** 6

**Strength And Weaknesses:**

Strengths:
- The proposed approach is novel.
- The performance of the approach is quite good.
- The ablations are very informative.

Weaknesses
- The method seems general enough to apply to all few-shot learning systems. Why restrict to the cross-domain setting?


**Summary Of The Paper:**

This paper proposes to perform few-shot recognition, and introduces two innovations. The first is that instead of averaging support image features to get a prototype, the proposed approach meta-learns a model that produces a prototype from concatenated features. The second innovation is to perform self-training with weighted moving averaging on the target domain.

**Summary Of The Review:**

I think this is a reasonable paper and worth accepting, but I would like to see how it does on the traditional FSL benchmarks.

---

> ### Author Response · Authors · 2022-11-18
> **Response to Reviewer TtNB**
>
> Thank you for acknowledging the novelty of our approach.
>
> * **The method seems general enough to apply to all few-shot learning systems. Why restrict to the cross-domain setting?**
>
> The cross-domain setting is much more challenging than the in-domain setting. The major challenge faced by cross-domain settings compared to in-domain settings is the domain shift problem. Our proposed method is specifically designed for the cross-domain setting. By employing learnable class prototypes and exploiting unlabeled data, the proposed approach can enable learning to learn and better adaptivity to bridge the gap between the source and target domain. Specifically, our proposed PCN is meta-learned on the source domain to perceive more discriminative and representative class prototypes from the given inputs, which can guide the model to better adapt to the target domain. Moreover, by exploiting unlabeled data and prototype regularization with WMA self-training, our proposed method can effectively mitigate the domain shift and adapt to the target domain.

---

### Decision · Program_Chairs · 2023-01-20

**Decision:**

Reject

**Justification For Why Not Higher Score:**

The novelty of this work is a bit limited.

**Justification For Why Not Lower Score:**

N/A

**Metareview: Summary, Strengths And Weaknesses:**

The article addresses the problem of cross-domain few-shot learning. It learns the class prototypes from a parametric prototype calculator network. The class prototypes are learned from concatenated feature vectors of the support instances to ensure inter-class discriminability and intra-class cohesion. The proposed approach is shown to be effective for cross-domain few-shot learning. The proposed method is tested on eight datasets, and slightly improved results are obtained on some datasets. It's more complete than some previous works.

However, the reviewers point out numerous issues such as the lack of motivation in the design choices, limited novelty, not clear contributions, many hyperparameters challenging to adjust for other tasks, lack of comparison with different cross-domain settings, the little contribution of knowledge for few-shot and cross-domain few-shot learning field, marginal or negative improvements on most of the datasets.

Accordingly, I suggest rejecting this article.